

# Gravity waves generated by the high graupel/hail loading through buoyancy oscillations in an overshooting hailstorm

**Xin Guo[1,3], Xueliang Guo[2], and Danhong Fu[2]**

[1]Beijing Weather Modification Center, Beijing, China
[2]Institute of Atmospheric Physics, Chinese Academy of Sciences, Beijing, China
[3]School of Atmospheric Sciences, Nanjing University, Nanjing, China

*Correspondence to*: X. L. Guo (guoxl@mail.iap.ac.cn), D. H. Fu (fudanhong@mail.iap.ac.cn)

**Abstract.** The convectively generated gravity waves (GWs) have important contributions on the stratospheric and mesospheric momentum and energy budget, and chemical composition, however, large uncertainties still remain about wave source properties and the associated wave-generated mechanisms. The formation mechanism and significant impacts of downward propagating GWs generated by a continental overshooting hailstorm occurred on 19 June 2017 in Beijing in the mid-latitude are reported in this study based on radar observations and simulated results from a three-dimensional cloud model with hail-bin microphysics. It is found that the overshooting storm penetrates the tropopause and enters the lower stratosphere in the mature stage. After the mature stage, the continuous descending process of the upper-level high graupel/hail loading causes the breaking of equilibrium between the buoyancy force and hydrometeor loading established in the mature stage and induces a restoring force of buoyancy, as well as buoyancy oscillations that excite downward propagating GWs. The GWs have a duration of about 20 min and the estimated wavelength of about 3-4 km. The downward propagating GWs not only result in the storm updraft splitting quickly, and significantly change the storm morphology and evolution, but also form the upward propagating GWs through surface reflection process, and induce strong vertical fluctuations in temperature and vertical velocity, and significantly change the dynamic and thermodynamic structure in the lower stratosphere.



## 1  Introduction

Atmospheric gravity waves (GWs) excited by deep convection have long been focused (e.g., Pierce and Coroniti,1966; Stull,1976; Fovell et al., 1992; Alexander et al., 1995; Piani et al., 2000; Horinouchi et al., 2002; Snively and Pasko, 2003; Müller et al., 2018), since convectively generated GWs have been found to have significant contributions to the momentum and energy budget (Fritts and Alexander,2003), and water vapor and tracers transport (Wang et al., 2002; Luderer et al.,2007) in the troposphere-to-stratosphere transport (TST), and even to the mesospheric chemical composition (Garcia and Solomon 1985).

The momentum flux associated with atmospheric GWs is observed much larger than those of Kelvin waves and Rossby-gravity waves (Sato and Dunkerton, 1997), and is an important driving forcing for some climate systems such as the quasi-biennial oscillation (QBO) (Alexander and Holton, 1997; Piani et al., 2000; Piani and Durran,2001; Baldwin et al., 2001; Beres et al., 2002).

The properties of atmospheric GWs are described based on the fluctuations of vertical wind and temperature profiles observed by lidar and radar (Larsen et al., 1982; Smith et al., 1985; Tsuda et al., 1989, 1994; Sato, 1992), and the generation mechanisms have been intensively investigated (Weinstock, 1985; Dewan and Good, 1986; Smith, 1987; Hines, 1991; Sato and Yamada, 1994; Warner and McIntyre, 1996; Nicholls and Pielke, 2000). The wave spectra, number and frequency of GWs have been also investigated (Lane and Moncrieff, 2008) and summarized (Gardner et al., 1993). The propagation and breaking of quasi-monochromatic small-scale GWs induced by thunderstorm activity were found to be closely associated with the observed airglow at the altitudes through the upper mesosphere and lower thermosphere (Snively and Pasko, 2003).

Atmospheric GWs can be generated by many sources, such as convection, wind shear, jet streams, frontal systems and topography, as well as pyro-cumulonimbus clouds (pyroCbs) induced by large forest fires (Luderer et al.,2007). Three main mechanisms for the convectively generated GWs have been proposed. One is referred to the thermal forcing mechanism, in which, the GWs are generated by convective



clouds through the latent heat release (Holton,1973,2002; Salby and Garcia, 1987;
Alexander et al., 1995; Mclandress et al., 2000; Fritts and Alexander, 2003). So that
the latent heating profile in convective storms determines the properties of GWs. Two
other mechanisms are referred to mechanical oscillation, such as the updraft
oscillation (Clark et al.,1986; Fovell et al., 1992; Alexander et al.,1995) and transient
mountain effect. However, the mechanisms for GWs generation by thermal forcing
and updraft oscillation are not easily separated since they are intrinsically coupled in
convective clouds. Lane et al. (2001) modeled GWs in maritime sea-breeze
convection and indicated that the mechanical oscillation mechanism was dominant in
GWs generation, while Song et al. (2003) suggested that the mechanical oscillation
and thermal forcing mechanisms had comparable magnitudes in GWs generation.
Numerical models have become an important role in investigation of atmospheric
GWs from single convective cloud models (Alexander et al.,1995; Fovell et al., 1992;
Lane et al.,2001) to General Circulation Models (GCMs) with convection
parameterization and convection-permitting schemes (Liu et al., 2014; Holt et al.,
2016; Müller et al., 2018).
Most of previous studies have focused on the convectively generated GWs from
thermal and mechanical oscillations of deep tropical convection and their influences
on the stratospheric atmosphere. The relevant studies on the GWs generated by
continental overshooting convection and their influences on the structure and
evolution of storms, as well as the stratospheric atmosphere remain unclear, merit
further investigation. Since the continent is the main region for human activity,
understanding how GWs generated by continental storms influence the structure and
evolution of storms, and the stratospheric atmosphere could be significant in storm
tracking and forecasting, as well as transport of momentum, energy and pollution
from the low troposphere to the upper atmosphere. The mountain-generated GWs
under certain meteorological conditions have been found to have important roles in
clouds and precipitation, as well as aerosol-cloud-precipitation interactions in
northern China (Guo et al., 2013, 2017). In this study, the properties and generation
mechanism of GWs, as well as the influences on both the storm itself and the



stratospheric atmosphere for a continental overshooting hailstorm occurred on 19 June
2017 are reported.

**2   Methods**
**2.1 Data**
The radar data observed by an operational SA-band Doppler radar located in the south
suburban of Beijing city are used to obtain the structure and evolution of the
GWs-generated overshooting storm. The radar data are also used to validate the
modeled storm. The sounding data from Beijing Meteorological station are used to
obtain the environmental conditions for the storm.
**2.2 The model**
A three-dimensional fully compressible nonhydrostatic cloud model with hail-bin
microphysics is employed to investigate the GWs properties, generation mechanism
and the effects in this study (Guo and Huang, 2002). The formation, growth and
conversion processes of cloud water, rainwater, cloud ice, snow and graupel/hail are
included in the model. The Kessler-type scheme is used for the warm microphysical
process (Kessler, 1969). The graupel/hail is categorized into 21 size bins ranging from
100 μm to nearly 7 cm in diameter. The model domain is on a standard spatially
staggered mesh system. The time-splitting integration technique is used to treat
high-frequency acoustic term (Klemp and Wilhelmson,1978). The large integration
time step is 5 s, while small time step is 0.25 s. The spatial difference terms are of
second-order accuracy except for the advection term that has fourth-order accuracy.
All other derivatives are evaluated with second-order centered differences. The
radiation lateral boundary condition is applied and top boundary is rigid. A Rayleigh
friction zone is used to absorb vertically propagating gravity waves near the top of the
domain. The model uses a first-order closure for subgrid turbulence and a diagnostic
surface boundary layer based on the Monin–Obukhov similarity theory.
The single sounding at 20:00 BST (Beijing Standard Time, the same hereafter) on
June 19, 2017 in Beijing is used to initiate the simulation. A thermal bubble located in
the central domain with a horizontal distance of 8 km and vertical distance of 4 km is





used for convection initiation in the model, and the maximum temperature
perturbation in the central bubble is 1.5℃. The total integration time is 80 min. The
domain size is 35 km in horizontal with a resolution of 1 km and 18.5 km in vertical
with a resolution of 0.5 km.

**3 Results**
**3.1 Environmental conditions**
The overshooting storm happened in the late afternoon on 19 June 2017 in the
northwestern Beijing city in the mid-latitude when a deep cold trough passed through
the city. The Beijing city was just located in the bottom of the trough with a strong
wind shear. An isolated convection was initially formed in the northeastern mountain
region of Beijing city, and developed as a severe overshooting hailstorm in 30 minutes.
The storm experienced multiple splitting processes, and produced rainfall over 50 mm
and hailstones of about 2.5 cm in diameter. The storm lasted for more than 2 hours.
Radar observations show that the overshooting storm top penetrated the tropopause
(~12 km) and reached up to 16 km in the lower stratosphere, and cloud base was
located around 20℃, indicating that the storm was a severe overshooting storm with a
warm cloud base and favorable for hail formation. The level for zero temperature was
around 3.7 km.
Fig.1 is the sounding profiles of temperature, dewpoint temperature and relative
humidity at 20:00 on June 19, 2017 in Beijing meteorological station. It indicates that
the atmospheric layer was relatively dry and the tropopause was located at around 12
km. The Convective Available Potential Energy (CAPE) and Convection Inhibition
(CIN) were 602 J/kg and 94 J/kg, respectively. The wind shear at 0-6 km was 19 m/s
with a southwesterly warm moist advection at the low-level and
northwesterly/westerly cold air advection at the high-level. Therefore, the atmosphere
had a potential unstable condition for convection initiation and development.
The hodograph exhibited a clockwise-turning from southwesterly winds near the
surface to northwesterly winds at approximately 5 km, and almost unidirectional
westerly winds at above 5 km. The low-level clockwise-turning hodograph is



favorable not only for right-moving storm splitting (Klemp and Wilhelmson, 1978),
but also for some long-lived, left moving storms (Grasso and Hilgendorf, 2001). The
Environmental Helicity (EH) and Storm Relative Environmental Helicity (SREH) in
this study were 73 and 30 J kg$^{-1}$, respectively, indicating that there was a relative weak
"helical" updraft (Johns and Doswell, 1992; Droegemeier et al. 1993).

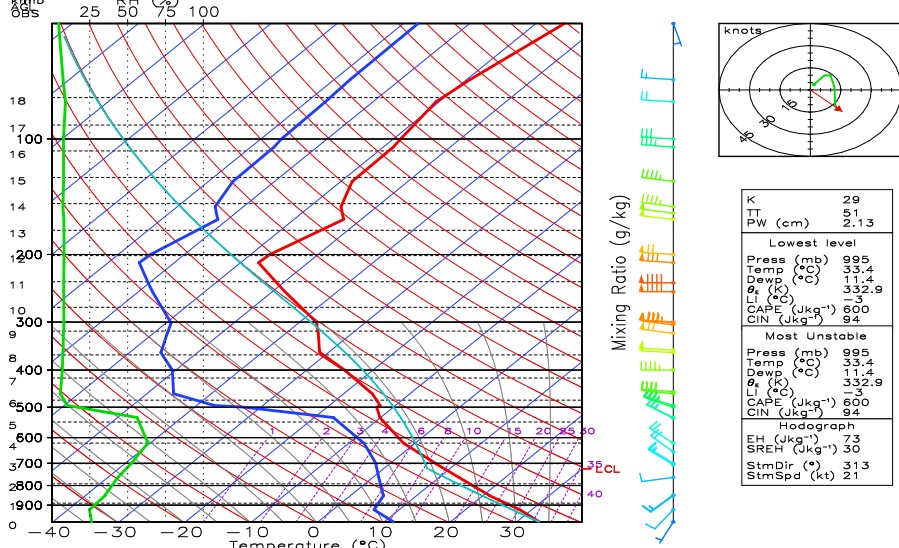


Fig.1. Profiles of temperature (red), dewpoint temperature (blue) and relative humidity (green) in
the Skew T-log P diagram at 20:00 on 19 June 2017 at Beijing Meteorological Station (39.8° N,
116.5° E). The winds, hodograph and environmental conditions are given on the right panel.

### 3.2 Observed and modeled properties of the storm

To understand the properties of the GWs-generated storm, the observed composite
radar reflectivity and corresponding vertical cross sections for the storm are shown in
Fig.2. At 19:30, a strong convection had already formed in the northeastern mountain
region of Beijing (Fig.2a$_1$). The corresponding vertical cross section in Fig.2a$_2$ shows
that the convection top was located at nearly 14 km and the maximum reflectivity was
40 dBZ, indicating that the storm had penetrated the tropopause (~12 km) and entered
the lower stratospheric layer. The storm was considered as a potential hailstorm at
19:42 since the high reflectivity was forming at the upper levels of the storm.



By 19:54, the storm entered the mature stage with the maximum reflectivity more than 60 dBZ and a pronounced leading stratiform region toward the northeast due to the influence of the strong southwesterly moist flow at the low- and mid-level (Fig.2b$_1$). Meanwhile, the storm also had an apparent development and extension toward the southeast. The vertical cross section in x-z in Fig.2b$_2$ shows that the storm top was more than 14 km, a large area shows as an overshooting structure although the storm top is relatively flat. Two high reflectivity cores with more than 50 dBZ were located at the height of 6-9 km and the reflectivity top with 40 dBZ reached up to 14 km, indicating that the graupel/hail was forming at the upper levels and the storm would produce hailfall soon.

At 20:06, the high reflectivity in the storm had an apparent development and expansion toward the east (Fig.2c$_1$). The southeastern extension of high reflectivity was also obvious. The striking phenomenon is that the upper-level reflectivity had an apparent V-shaped splitting structure, which should be closely related to the high reflectivity descending process (Fig.2c$_2$). Meanwhile, the storm top experienced an explosive growth for about 2 km from 14 to 16 km and the overshooting structure became pronounced. The simulated results in the next section will show that the downward propagating GWs are generated at this stage.

At 20:12, the development and extension of high reflectivity toward the east became more obvious than that toward the southeast (Fig.2d$_1$), suggesting that the storm splitting in the west-east direction was faster than that in the south-north direction, although the developments toward the both directions were initiated almost at the same time. The V-shaped reflectivity splitting structure became more obvious on the eastern flank of the storm due to the further descending of the upper-level high reflectivity (Fig.2d$_2$). Corresponding to the high-reflectivity descending process, the cloud top was decreased to 14 km.

By 20:18, the mid- and upper-level high reflectivity in the storm had already split in the west-east direction (Fig.2e$_1$, e$_2$), indicating the main updraft of the storm had split into two independent updrafts. After 20:18, the storm development and expansion toward the southeast tended to enhance and became more pronounced. The





splitting in the south-north was quite similar to that in the west-east direction, but the
reflectivity splitting occurred in the central storm. Since the paper mainly focuses on
the properties of the storm and associated generation mechanism of GWs, the splitting
mechanism is out of the scope of this study.

206        As stated above, the observed storm had two pronounced features, one was that the

storm top penetrated the tropopause and entered the lower stratosphere in the mature.
The other was the V-shaped reflectivity splitting structure associated with the
descending of the upper-level high reflectivity, and the accompanied explosive growth
of storm top and the overshooting structure after the mature stage.

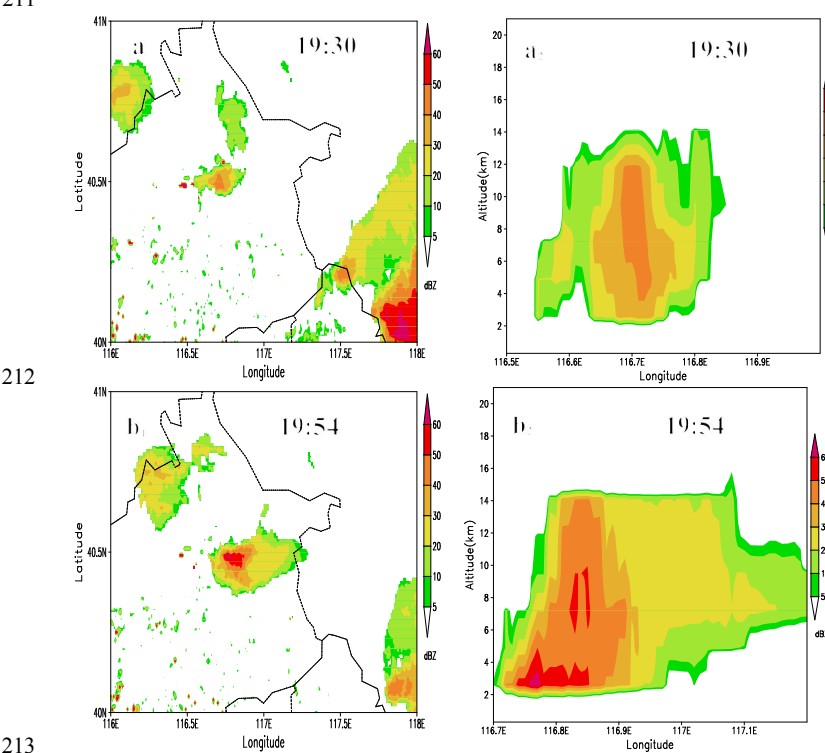





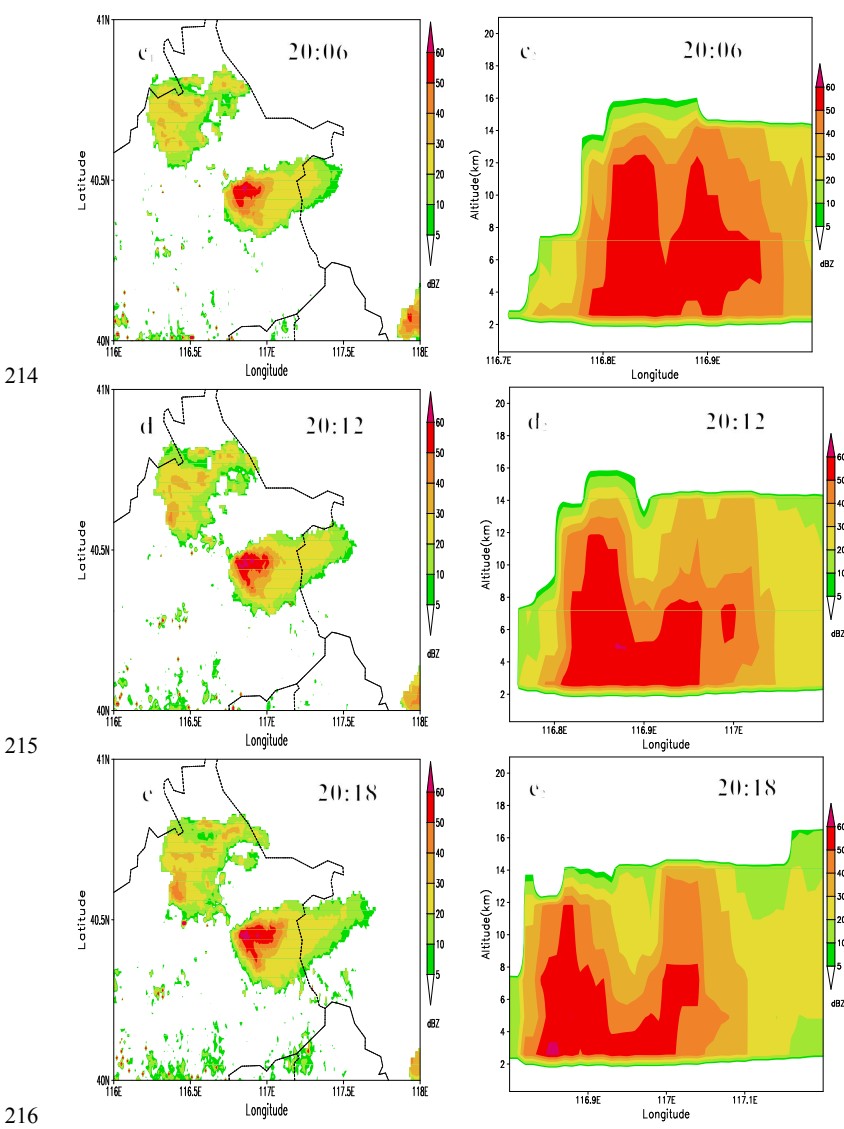




Fig.2. Observed composite radar reflectivity ($a_1$-$e_1$) and corresponding vertical cross sections
along the high echo cores in the west-east direction ($a_2$-$e_2$) at (a) 19:30, (b) 19:54, (c) 20:06, (d)
20:12, and (e) 20:18 on 19 June 2017 in Beijing.

To compare with the observed storm, the temporal evolution of the simulated
mixing ratio of total hydrometeors for the modeled storm in the x-z (west-east) cross
section is displayed in Fig.3. At 10 min, which is roughly corresponding to the
observed time 20:10 since the model simulation is initiated with the sounding data at
20:00 (the starting time for operational sounding is 19:15). A vigorous convection is



formed with the cloud-top height of 10 km, and the maximum mixing ratio of total
hydrometeors reaches more than 15 g/kg (Fig.3a). By 12 min (Fig.3b), the modeled
storm has the cloud-top height of 14 km and the maximum mixing ratio of 20 g/kg,
indicating that the storm enters the mature stage with a very high loading of
hydrometeors (graupel/hail particles, see Fig.4$a_1$-$a_3$) at the upper levels. The
overshooting storm top penetrates the tropopause (~12 km) and enters the height of 14
km in the lower stratosphere, which is well consistent with radar observations. The
modeled storm has the maximum updraft of about 60 m/s in the mature stage and
downdraft of about -35 m/s.
At 14 min (Fig.3c), the upper-level high total hydrometeor mixing ratio
significantly decreases from 20 to 15 g/kg due to the strong descending process of
upper-level graupel/hail on the eastern flank of the storm. Meanwhile, the modeled
storm top increases from 14 to about 16 km. The overshooting storm structure become
more pronounced. All modeled features are well consistent with those observed by
radar (Fig.2$b_2$, $c_2$).
By 16 min (Fig.3d), the total hydrometeor mixing ratio decreases from 15 to 10
g/kg and the continuous descending of upper-level graupel/hail further strengthens the
cloud-top height. At 18 min (Fig.3e), the total hydrometeor distribution tends to have
an obvious V-shaped splitting structure, which is also consistent with the V-shaped
reflectivity splitting structure observed by radar, indicating that the V-shaped splitting
structure is closely associated with the descending of precipitating hydrometeors. The
area with the mixing ratio of 10 g/kg decrease significantly, indicating that the
cloud-top height tends to decrease in the region with apparent descending
hydrometeors.
By 20 min (Fig.3f), the further descending of precipitating hydrometeors cause the
maximum mixing ratio to decrease from 10 to 5 g/kg in the almost whole storm, and
the V-shaped splitting structure of hydrometeor mixing ratio descends to the lower
levels. The cloud-top decreases to about 14 km.
As described above, the properties of the modeled storm and descending processes
of upper-level precipitating hydrometeors are generally consistent with radar





observations. It is shown that the storm penetrates the tropopause and enter the lower
stratosphere in the mature stage. The strong descending of precipitating hydrometers
causes more pronounced overshooting structure. The continuous descending of
precipitating hydrometeors can induce an apparent V-shaped splitting structure as that
observed by radar after the mature stage.



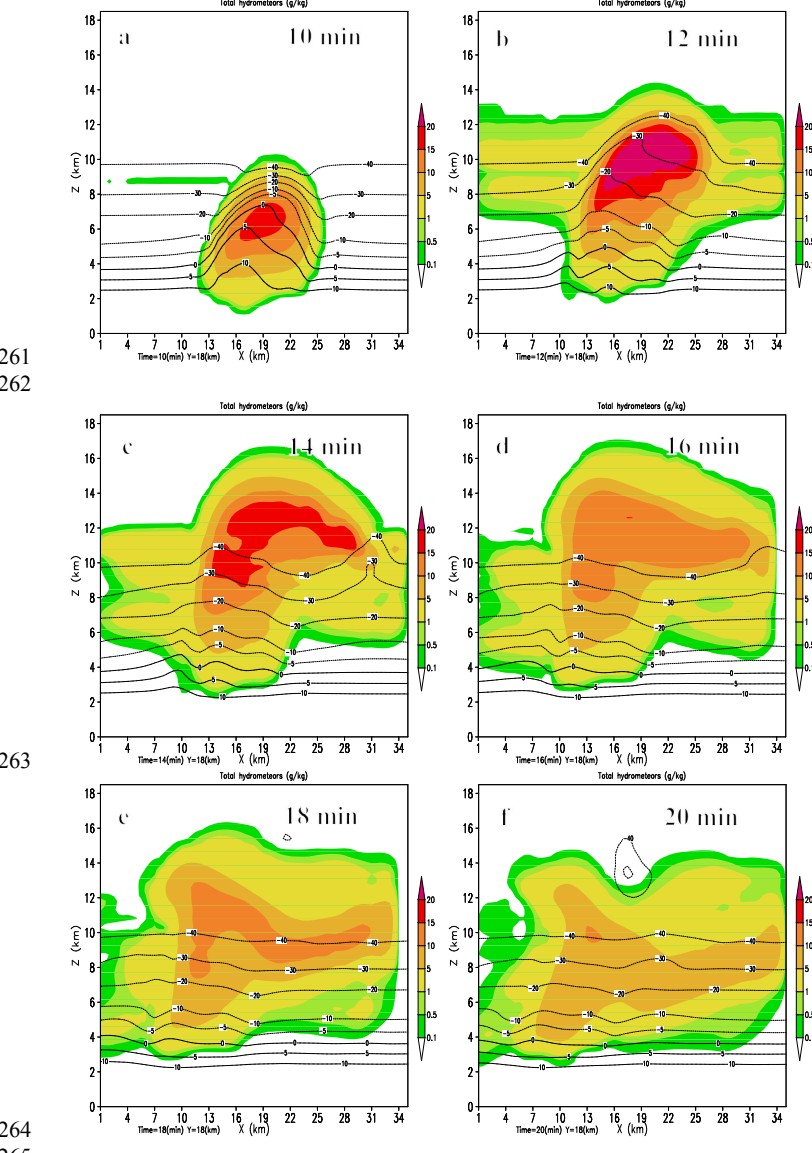

Fig.3. Temporal evolution of the simulated mixing ratio of total hydrometeors in the x-z





(west-east) cross section for the modeled storm from 10 to 20 min on 19 June, 2017. The
horizontal solid and dashed lines in figures are environmental positive and negative temperatures,
respectively.

**3.3 The generation mechanism and relevant properties for the GWs**
To investigate the generation mechanism and relevant properties of the GWs, the
temporal evolution of graupel/hail mixing ratio, pressure perturbation, temperature
perturbation and vertical velocity from 14 to 18 min in the x-z cross section are shown
in Fig.4.

As stated above, the overshooting severe storm is formed when the simulated storm

is in the mature stage at 12 min. In this stage, the storm reaches its maximum updraft
of 60 m/s and an equilibrium between the buoyancy force and hydrometeor loading is
established. As long as the updraft cannot further sustain the upper-level high
hydrometeor loading, the high precipitating hydrometeors may break the equilibrium
and descend after the mature stage.

At 14 min, the upper levels are dominated by high graupel/hail loading with the

maximum mixing ratio of more than 15 g/kg (Fig.4$a_1$), indicating the upper-level high
reflectivity observed by radar and hydrometeors simulated by the model are due to the
graupel/hail particles. The corresponding pressure perturbation distribution in Fig.4$b_1$
shows that there is a strong positive pressure perturbation of more than +3 hPa at the
upper levels of 8-15 km on the western flank of the storm, which is related to the
strong updraft and latent heating on the flank (Fig.4$d_1$). The high graupel/hail loading
is just located in the region of strong positive pressure perturbations, so that the
fluctuation of the high graupel/hail loading may significantly change the pressure
perturbation. A tilting and relatively uniform positive pressure perturbation with +1
hPa penetrates the middle and lower levels and corresponds an obvious wavelike
temperature perturbation (Fig.4$c_1$), indicating that downward propagating GWs have
already occurred at this stage since the cloud-top height has an apparent upward
extension. The upper-level high negative temperature perturbation region is closely
associated with the strong outflow at the cloud top. A small negative temperature
perturbation region located just below the high negative temperature region should be
caused by the downward propagating GWs. The positive temperature area located on
the eastern flank is due to the adiabatic warming of downdraft. The propagating GWs
cannot be seen clearly in the pressure perturbation due primarily to that strong
background pressure perturbations offset the effect induced by the GWs. Since there
is no apparent change in pressure perturbation, the vertical velocity in the area is still
dominated by updraft (Fig.4d$_1$).
By 16 min, the maximum graupel/hail mixing ratio decreases to be lower than 15
g/kg due to the apparent descending of graupel/hail particles at the upper levels on the
eastern flank of the storm (Fig.4a$_2$). In response to the significant decrease of the
upper-level graupel/hail loading, the equilibrium between the buoyance force and
hydrometeor loading is destructed and a strong restoring force of buoyancy is
produced in the stratosphere. The formation of the restoring force of buoyancy causes
the overshooting structure to be more prominent. The buoyancy oscillations induced
by the continuous descending of the upper-level graupel/hail in the overshooting
storm induce a pronounced downward propagating GWs, which is can be clearly seen
in pressure perturbation (Fig.3b$_2$). In the pressure perturbation distribution, the
positive pressure perturbation is generally not as obvious as the negative pressure
perturbation, this is because that when the downward propagating GWs penetrate the
high-pressure region dominated by the updraft, the positive pressure perturbation
induced by the GWs should be much smaller than that induced by the updraft of the
storm.
The estimated wavelength of the GWs is around 3-4 km. Accompanying with the
strengthening downward propagating GWs, the temperature perturbation is further
enhanced (Fig.4c$_2$). As a result, the main updraft in the storm is split into two
independent updrafts as shown in Fig.4d$_2$, indicating that the rapid updraft splitting is
closely associated with the downward propagating GWs generated by buoyancy
oscillations induced by the descending of the upper-level high graupel/hail. The weak
downdrafts of -1~-3 m/s are distributed along the path of downward propagating GWs,
indicating the downward momentum transport associated with the downward



propagating GWs can damage the updraft of the storm and induce the storm splitting
rapidly. The strong compensating subsidence with the magnitude of -15 m/s in the
stratosphere should be primarily induced by the downward propagating GWs
(Bretherton and Smolarkiewicz,1989), and the descending of graupel/hail particles.
At 18 min, the continuous descending of graupel/hail on the eastern flank of the
storm significantly decreases the upper-level graupel/hail loading (Fig.4$a_3$).
Meanwhile, the descending of graupel/hail particles tends to shift toward the west and
induce a westward shifting of downward propagating GWs (Fig.4$b_3$). An interesting
phenomenon at this stage is that the GWs also have an apparent upward propagation
to the lower stratospheric layer. Since the downward propagating GWs are generated
through buoyancy oscillations induced by the descending of the upper-level
graupel/hail in the stratospheric layer as shown in Fig.4$b_2$, the upward propagating
GWs should be induced by the surface-reflected GWs when the downward
propagating GWs reach to the surface as proposed by Kim et al. (2012). It will be
seen in the following section that the upward propagating GWs can generate a strong
horizontally propagating GWs in the lower stratospheric layer. In fact, the temperature
perturbation has already shown an obvious horizontal wavelike distribution in the
layers above the tropopause (~12 km) (Fig.3$c_3$). The temperature perturbation below
the tropopause tends to weaken due to the effect of the upward propagating GWs.
With the descending of the upper-level graupel/hail and the formation of strong
downdraft, the storm tends to weaken significantly (Fig.4$d_3$). Note that the vertical
velocity in the stratospheric layer also tends to have a wavelike perturbation.
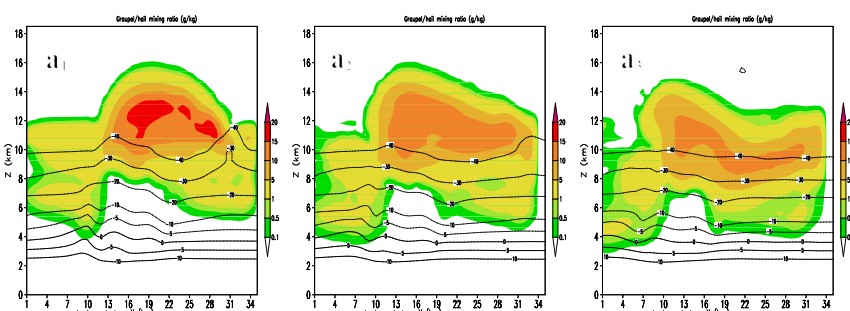

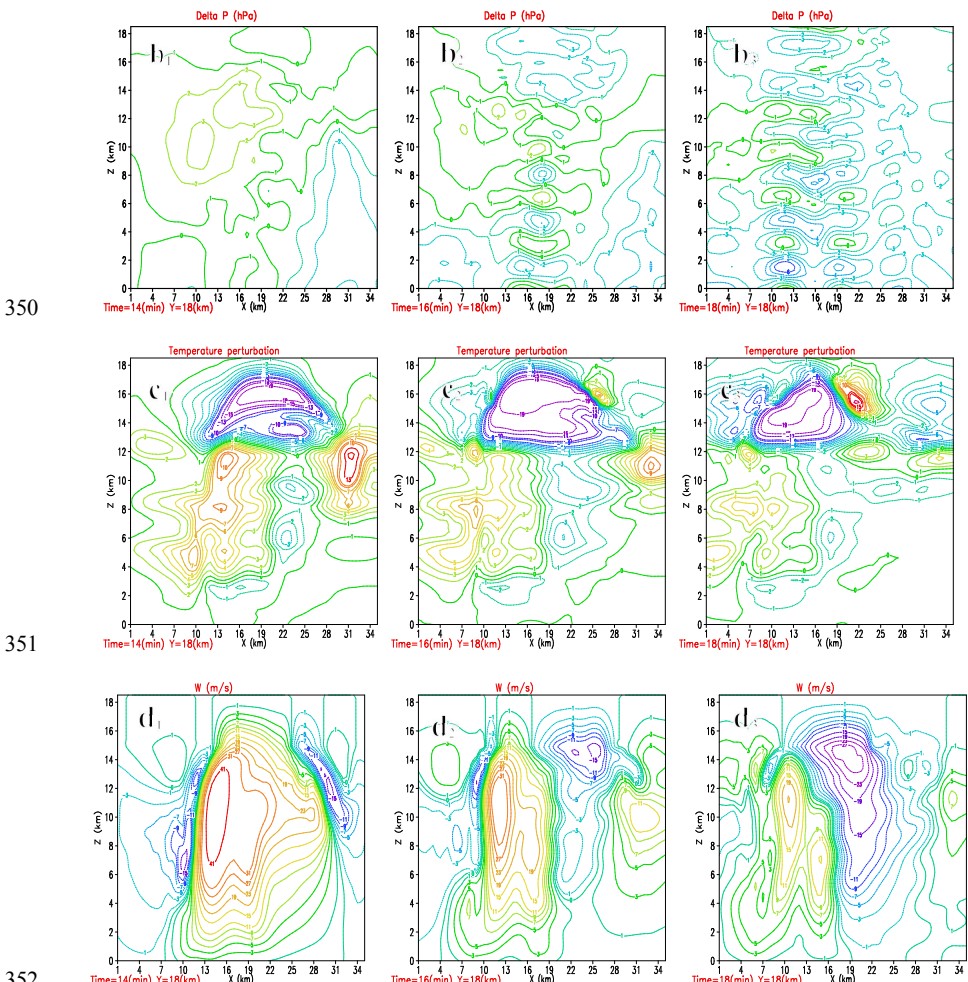

Fig.4. Temporal evolution of (a₁-a₃) graupel/hail mixing ratio (g/kg), (b₁-b₃) pressure perturbation (hPa), (c₁-c₃) temperature perturbation, and (d₁-d₃) vertical velocity (m/s) in x-z cross section at y=18 km from 14 to 18 min. The horizontal solid and dashed lines in (a) are environmental positive and negative temperatures, respectively. The updraft is solid lines and downdraft is dashed lines in (c).

As described above, the storm penetrates the tropopause and enters the lower stratospheric layer and forms an overshooting severe storm in the mature stage. The descending of the upper-level high graupel/hail loading in the overshooting storm breaks the equilibrium between the buoyancy force and hydrometeor loading



established in the mature and induce a strong restoring force of buoyancy. The
continuous descending processes produce buoyancy oscillations that excite the
downward propagating GWs. The downward propagating GWs can produce apparent
downward perturbations in pressure, temperature and vertical velocity, resulting in a
rapid storm splitting. The initially induced downdraft by the GWs is around -1~ -3
m/s. The GWs generated by the overshooting severe storm have a duration of about
20 minutes and an estimated wavelength of about 3-4 km. Although the generation
mechanism of the GWs in this study is different from those proposed by previous
studies, the relevant features of the GWs are generally similar to convectively
generated GWs through other mechanisms (e.g., Larsen et al.1982; Nolan and Zhang,
2017; Jewtoukoff et al. 2013). Larsen et al. (1982) observed GWs generated by
afternoon thunderstorms with a vertically-pointing 430 MHz radar and found that
when the cloud-top height reached the tropopause, gravity-wave oscillations in the
vertical velocity above the tropopause would develop, with an amplitude of 2 m/s, and
period of close to 6 min. The aircraft measurements by Nolan and Zhang (2017)
indicated that the GWs have radial wavelengths of 2–10 km and vertical velocity
magnitudes from 0.1 to 1.0 m/s. Jewtoukoff et al. (2013) reported the GWs near a
tropical cyclone with wavelengths of around 1 km observed by a balloon at 19 km
altitude. In addition, the upward propagating GWs induced by the surface reflection
are also obvious in the simulation and will be further discussed in the next section.

**384    3.4 The influences of the surface-reflected GWs on the stratosphere**

As shown above, both the downward and upward propagating GWs are formed
through buoyancy oscillations and surface reflections in the continental overshooting
hailstorm. It is shown that the downward momentum transport associated with the
downward propagating GWs can induce the rapid updraft splitting and change the
storm morphology and evolution. One of important issues is that whether the
surface-reflected upward propagating GWs can also affect the stratospheric
atmosphere through the upward momentum and energy transport as proposed by
previous studies (e.g., Alexander and Holton, 1997; Piani et al., 2000; Baldwin et al.,



2001; Beres et al., 2002; Fritts and Alexander,2003). To investigate this issue, the
subsequent evolution of cloud total hydrometeor, pressure and temperature
perturbations, and vertical velocity from 20 to 50 min for the simulated storm is
displayed in Fig. 5.
At 20 min, the upper-level mixing ratio of graupel/hail decreases to be less than
10 g/kg (Fig.5$a_1$). The Fig.5$b_1$ shows that the wavelike pressure perturbation induced
by the downward propagating GWs continues to shift toward the west due to the
westward shifting of descending process of graupel/hail. The pressure perturbation
tends to weaken due to the weakening of graupel/hail loading. It can be clearly seen
that the surface-reflected upward propagating GWs induce an obvious positive
pressure perturbation in the lower stratosphere (Fig.5$b_1$). This phenomenon is more
prominent in the temperature perturbation (Fig.5$c_1$). In the lower stratosphere, the
pronounced wavelike fluctuations in temperature perturbation can be clearly seen,
indicating that the momentum and energy transport associated with the upward
propagating GWs can enter the lower stratospheric layer and generate strong
horizontally propagating GWs as observed by aircraft (Nolan and Zhang (2017).
It should be noted here that the temperature distribution pattern with a warm
center surrounded a U-shaped or V-shaped cold region in the lower stratospheric layer
over the storm top is quite similar to those found in the pyroCbs (Luderer et al.,2007)
and intense thunderstorms (Wang et al., 2002) induced by GWs. However, the GWs in
the lower stratosphere in this study are generated by the surface reflection of
downward propagating GWs rather than that directly produced on the storm top. The
formation of the wavelike distribution in vertical velocity can be also seen in the
lower stratosphere, although the vertical velocity distribution is still dominated by
main updraft and the compensating subsidence.
By 30 min, the graupel/hail descends to the lower levels and some of them melt as
rainwater, so that high mixing ratio of hydrometeor presents at the surface (Fig.5$a_2$).
The cooling caused by both the melting and evaporating processes causes the pressure
at the near-surface to decrease significantly (Fig.5$b_2$). The small surface positive
pressure should be related to the cold downdraft. The surface-reflected upward





propagating GWs induce a new temperature fluctuation at the lower levels of the
stratosphere (Fig.5c$_2$). A strong cold pool with the thickness of 4 km is formed at the
near-surface layer with the minimum temperature of -15 ℃. The downdraft is
dominated in the cold pool (Fig.5d$_2$). The surface-reflected upward propagating GWs
also induce apparent fluctuations in vertical velocity in the stratosphere.
At 40 min, the cloud-top descends to the height below the tropopause and the
graupel/hail descending process has weakened significantly (Fig.5a$_3$). Since the
restoring force and buoyancy oscillations cannot be formed in the troposphere, the
downward propagating GWs cannot be also generated (Fig.5b$_3$), instead, the low-level
evaporative cooling produces a strong negative pressure perturbation in the
near-surface layers. The strong cold pool spreading cause the surrounding air to lift
and condense (Fig.5c$_3$). Meanwhile, it seems that the strong cold pool spreading also
generates the weak upward propagating GWs in the stable low layers, and induces
relatively weak horizontal temperature fluctuations in the low stratosphere. The
vertical velocity distribution in Fig.5d$_3$ shows that within the cold pool there is
downdraft while above the cold pool there is a weak updraft due to the lifting of
spreading outflow. In the stratosphere, there are horizontally propagating weak
fluctuations in vertical velocity with an amplitude of around 1-3 m/s. Therefore, the
strong cold pool spreading at the low levels could also generate upward propagating
GWs and induce the momentum and energy transport from the low tropospheric
levels to the upper stratospheric layers.
By 50 min, the convective cloud has evolved as a stratiform cloud (Fig.5a$_4$). Since
precipitation and the associated melting and evaporative cooling weaken significantly
and the downward cold airflow become dominant (Fig.5d$_4$), the near-surface layer is
dominated by positive pressure perturbation (Fig.5b$_4$). The spreading outflow induced
by the cold pool continuously lifts and condensates the air above it and form a weak
positive temperature perturbation (Fig.5c$_4$). The vertical velocity distribution in
Fig.5d$_4$ shows that there is a downdraft in the cold pool and weak uplifting velocity
above the cold pool. The horizontal GWs are no longer to propagate in the
stratosphere. Therefore, comparing with the GWs generated by buoyancy oscillations



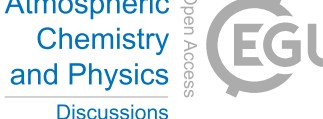

induced by the descending of the upper-level high graupel/hail, the GWs excited by
the strong spreading outflow of the cold pool are relatively weak and have less impact
on the stratosphere. However, the cold pool has an important role in maintaining the
subsequent clouds and precipitation through the lifting process.

Therefore, the surface-reflected upward propagating GWs have significant impacts

on the temperature and vertical velocity distributions in the stratosphere, indicating
that the GWs generated by the overshooting severe hailstorm not only influence the
storm morphology and evolution through downward propagating process, but also
significantly affect the stratospheric atmosphere through the surface-reflected upward
propagating process. The GWs excited by the strong spreading outflow of the cold
pool are relatively weak. But the spreading outflow of the cold pool has an important
role in maintaining the subsequent development of clouds and precipitation through
the lifting process. When the environmental air is under the unstable condition, the
lifting could induce convection and a longer duration of the storm. This property is
apparent in the splitting in the south-north direction (not shown).


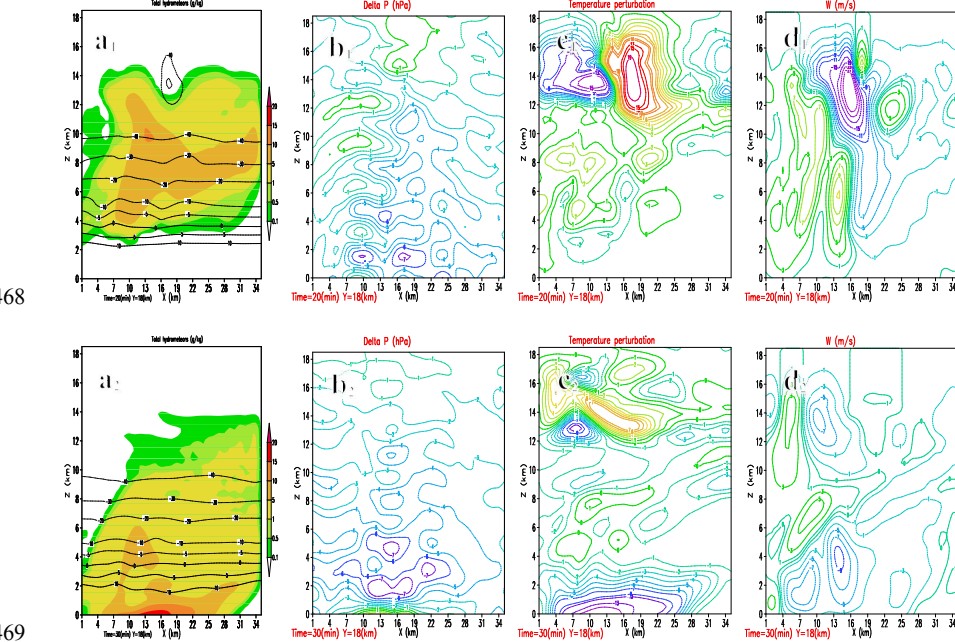

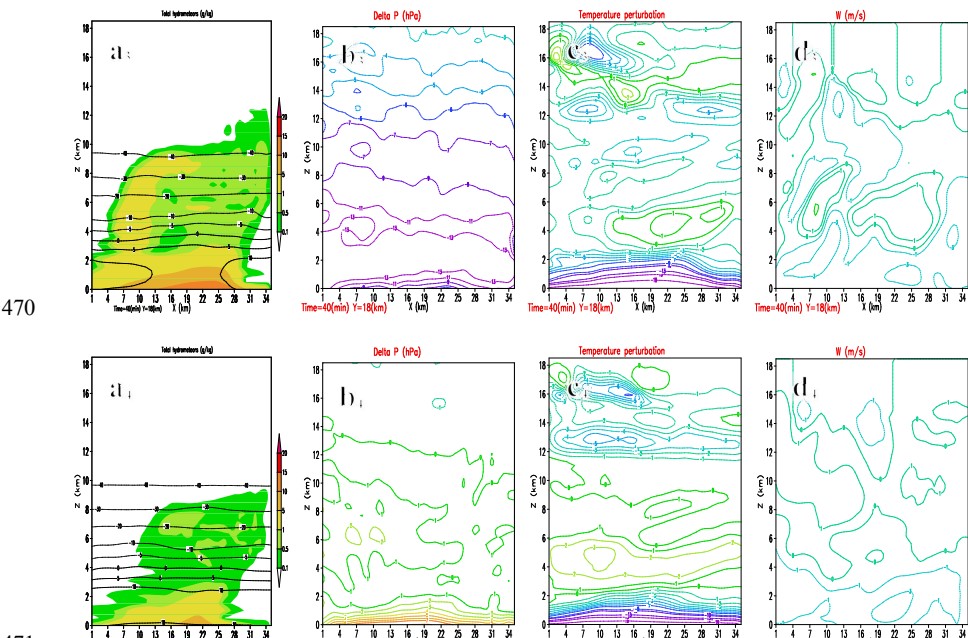



Fig.5. Vertical cross sections of ($a_1$-$a_4$) graupel/hail mixing ratio (g/kg), ($b_1$-$b_4$) pressure perturbation (hPa), ($c_1$-$c_4$) temperature perturbation, and ($d_1$-$d_4$) vertical velocity (m/s) in x-z at y=18 km from 20 to 50 min. The horizontal solid and dashed lines in (a) are environmental positive and negative temperatures, respectively. The cloud boundary is superimposed as a thick solid curve for 0.1 g/kg in (b). The updraft downdraft is solid lines and is dashed lines in (d).

## 4 Conclusions and discussion

The GWs generated by a continental overshooting hailstorm occurred on 19 June 2017 in Beijing in the mid-latitude are first reported in this study based on radar observations and modeled results. The main conclusions are summarized as follows.

The GWs-generated overshooting hailstorm has the maximum cloud-top height of over 16 km, updraft of 60 m/s and graupel/hail mixing ratio of over 20 g/kg in the mature stage. The storm penetrates the tropopause and enters the lower stratosphere and forms a typical overshooting storm. After the mature stage, the descending of the upper-level high graupel/hail loading causes the breaking of equilibrium between the buoyancy force and hydrometeor loading established in the mature stage, and induce a



strong restoring force of buoyancy. The continuous descending processes of the upper-level high graupel/hail loading produces buoyancy oscillations that excite downward propagating GWs. The GWs have the estimated wavelength of about 3-4 km and duration of about 20 min.

The momentum flux associated with the downward propagating GWs produces downdraft, and cause the main updraft splitting quickly in the storm, and significantly change the storm structure and evolution. The downdraft magnitude induced by the GWs is about -1~-3 m/s in the initial stage. The upward propagating GWs can be also formed through the surface reflection of the downward propagating GWs. The upward propagating GWs are trapped in the lower stratosphere and induce the large fluctuations in temperature and vertical velocity, causing significant influences on the dynamic and thermodynamic structure in the low stratosphere.

The generation mechanism of the GWs reported in this study is different from the convectively generated GWs mechanisms through mechanical, thermal and mountain forcing proposed by previous studies, since the convectively generated GWs through mechanical and thermal forcing mechanisms are closely associated with latent heating release and updraft fluctuation, and generally propagate upward with a restoring force of gravity, so that the GWs have significant contributions to the stratospheric momentum and energy budget (Fritts and Alexander, 2003), while the GWs reported in this study are excited by buoyancy oscillations caused by the continuous descending processes of graupel/hail in an overshooting hailstorm. The restoring force is buoyancy. The GWs propagate downward and have important impacts on the storm morphology and evolution, as well as lower stratosphere through the surface reflection process.

The properties of the GWs generated by the overshooting hailstorm in this study are generally consistent with radar and aircraft observations (Larsen et al.,1982; Jewtoukoff et al., 2013; Nolan and Zhang, 2017). The temperature distribution pattern with a warm center surrounded a U-shaped or V-shaped cold region in the lower stratospheric layer over the storm top is quite similar to that found in the pyroCbs (Luderer et al.,2007) and intense thunderstorms (Wang et al., 2002). Luderer et



al.(2007) proposed that small-scale mixing processes are strongly enhanced by the
formation and breaking of a stationary gravity wave induced by the overshoot.
However, the GWs in the lower stratosphere in this study are generated by the surface
reflection process of downward propagating GWs rather than that directly induced by
the storm overshoot. In addition, it should be noted that the overshooting hailstorm in
this study has the maximum updraft of 60 m/s, cloud-top height up to 16 km and
graupel/hail mixing ratio up to 20 g/kg in the mature stage, so that the strong
downward propagating GWs can be generated through buoyancy oscillations induced
by the continuous descending processes of the upper-level high graupel/hail loading.
Whether the GWs can be generated in general continental severe hailstorms through
these processes remain uncertain and needs further study in the future.

Data availability. Radar and sounding data used in this study are available from the National
Meteorological Information Center (NMIC), China Meteorological Administration (CMA),
website: http://www.nmic.cn/.

Author contributions. XLG conceptualized and designed the study.
XG, DF and XLG performed data analysis and numerical simulations.
XG and DF conducted formal analysis and wrote the manuscript.
All authors read and approved the final manuscript.

Competing interests. The authors declare that they have no conflict of interest.

*Acknowledgements.* This work was supported by the National Natural Science Foundation of
China (42105173) and the Second Tibetan Plateau Comprehensive Scientific Expedition
(2019QZKK0104).

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
