# Peer review of "Gravity waves generated by the high graupel/hail loading through buoyancy oscillations in an overshooting hailstorm"

_Atmospheric Chemistry and Physics, 2022_

## Author Comment (AC1)

General comments:

This paper reproduces the gravity waves generated by graupel/hail loading in a hailstorm, which were captured by radar observations, in a numerical model and examines their generation, propagation, and impact on the storm and stratosphere. The mechanism of gravity wave generation, in which the vertical equilibrium is disrupted by graupel/hail loading, looks new and interesting to the reviewer. Although the paper describes gravity waves based on temperature, pressure, and vertical wind perturbations, it is unclear which variations in the figures are corresponding to gravity waves for the most part, and many of them do not establish phase relationships among temperature, pressure, and vertical wind. In addition, four types of gravity waves (upward-propagating, downward propagating, reflected, and trapped) are described, but the authors do not specify on what basis they made such judgments. The figures are also difficult to read, which makes it difficult to judge whether the authors' claims are correct or not. For these reasons, we consider it appropriate to reject this paper.

**Reply**: We appreciate your important and detailed comments. The main concerns raised by the reviewer have two aspects: one is that which variations of temperature, pressure, and vertical wind perturbations are corresponding to gravity waves. The other is the basis for the proposed gravity waves (upward-, downward-, reflected and trapped). The reasons that cause such confusions are largely due to that the important properties obtained by this paper are not clearly indicated on the relevant figures and unclear descriptions in the text. Therefore, we have carefully considered all comments and revised the paper. Some apparent indications added on the figures may help readers to catch significant features obtained in this study. More detailed and clearer descriptions are added in the revised the manuscript. All figures are revised to be clearer based on the comments. Some important features relevant to gravity waves and their impacts are explained as following. Figures 1-7 are directly copied from the revised manuscript.

Fig.1 shows downward propagating gravity waves clearly indicated by symbols. It shows that the downward propagating waves indicate as wavelike property in pressure perturbations. The reason to attribute the waves as downward gravity waves is mainly based on two aspects: one is that the waves are closely corresponding to the descending of strong reflectivity in radar observation and high graupel/hail loading. The other aspect is that the downward waves produce strong downdraft and can be clearly shown in the z-t evolution of vertical velocity in Fig.5b. The descending of graupel/hail forces the air to be displaced downward and disrupts the air equilibrium state and the recovery buoyancy is formed to act against this in the stable lower stratosphere (about 14 km). Since the descending of graupel/hail initiates on the right flank of the storm and tends to shift toward the left, so that downward gravity waves also moves from the right to the left.

Fig.2 shows both the downward and upward gravity waves. The downward gravity waves are reflected by the surface and propagate upward and enter the upper levels. The reason for identifying the upward gravity waves is also based on two aspects: one is the wave propagating path and property. Since the upward waves propagate from the surface to the stratosphere, no additional source can generate this wave. The other aspect is that we can clearly see this in the z-t evolution of vertical velocity in Fig.5a. There is a strong updraft formed by the upward waves

after a strong downdraft is formed by the downward waves. These features cannot be explained by the general evolution and structure of a storm.

Fig.3 shows that the strong downdraft caused by the downward gravity waves split the main updraft into two parts. The reviewer questioned that the storm splitting does not occur. This is misled by the distribution of total mixing ratio. The storm splitting here refers to the updraft splitting rather than the whole storm splitting. The updraft splitting corresponds well with the high core of reflectivity rather than total mixing ratio. In order to avoid this confusion, we replace the total mixing ratio as the simulated reflectivity, which might be better to compare with observed reflectivity by radar as shown in Fig.7.

Fig. 4 shows the upper-level temperature perturbations induced by the upward propagating gravity waves, which can be well explained by Temporal-height (z-t) distributions of maximum and minimum temperature perturbations for the simulated storm shown in Fig.6.

Fig.5 shows the temporal evolution of vertical velocity for the simulated storm. It clearly shows that the updraft and downdraft marked by rectangular boxes cannot be explained by the normal evolution of vertical velocity in a storm. The abnormal downdraft occurs prior to the updraft and well corresponds to the downward gravity waves while the updraft is closely linked to the perturbation of updraft induced the trapped upward gravity waves in the stratosphere.

Fig.6 shows the temporal-height (z-t) distributions of maximum and minimum temperature perturbations for the simulated storm. It shows the positive and negative temperature perturbations marked by rectangular boxes re cannot be explained by normal temperature perturbations. The abnormal temperature perturbations are strongly related to the upward propagating gravity waves and associated momentum deposition and propagation in the lower stratosphere.

Fig.7 The modeled reflectivity shows an obvious reflectivity splitting on the eastern flank of the storm.

[Figure]

Fig.1 downward propagating waves

[Figure]

Fig.2 Downward and upward propagating gravity waves

[Figure]

Fig.3 Storm updraft splitting induced the downward propagating waves.

[Figure]

Fig.4 Upper-level temperature perturbations induced by upward propagating gravity waves trapped in the lower stratosphere.

[Figure]

Fig. 5 Temporal-height distribution of (a) maximum updraft (m/s) and (b) downdraft (m/s) for the simulated storm

[Figure]

[Figure]

Fig.6. Temporal-height (z-t) distributions of (a) maximum temperature (℃) and (b) minimum temperature (℃) for the simulated storm on 19 June 2017.

[Figure]

Fig.7 Modeled reflectivity shows the reflectivity splitting on the eastern flank of the storm

Specific comments:

1. Introduction

There is too little information on past studies on convective GWs. For example, it should be described how convective GWs affect the stratosphere and/or the source storms more specifically. It should also be described how graupel/hail, the subject of this study, is or is not addressed in the past studies and what might change by considering graupel/hail.

**Reply:** accepted. More information has been added in the revised paper.

2.1 Data

Basic specifications of the radar such as station latitude/longitude, temporal/vertical resolutions, observation range in the vertical and horizontal, etc. are not described. There is no reference.

**Reply:** accepted. The information has been added.

2.2 The model

Is the method of giving the thermal bubble, its size and amplitude optimized? Are different values

of each parameter tested?

**Reply:** This has been done in many years ago. We have concluded that size and amplitude of the thermal bubble can change the initiation time of a convection, but cannot change the overall evolution and structure of a storm, since the thermal bubble is only used in the initial state and help to lift the air parcel to the Free Convection Level (FCL).

Description of GWs

- L. 291-295, 296-298

Why can you say that the pressure and temperature perturbations are due to downward-propagating GWs? I cannot catch which part of the figure is the downward-propagating GWs. Please show the t-z section.

**Reply:** this is given and explained above. Fig.6 shows the z-t section of maximum and minimum temperature perturbations. The part marked by a rectangle in upper figure is that induced by the upward gravity waves trapped in the stratosphere, which is well consistent with upward propagating gravity waves in time and space. The part marked by a rectangle in the lower figure is that induced by the downward gravity waves. For a non-overshooting or general storm, this feature cannot occur. The near-surface temperature perturbation is induced by downdraft and cooling processes, which forms the cold pool.

- L. 299-303

The previous sentences state that pressure perturbation is due to GWs, but is the background pressure perturbations here different from that? If so, what is it due to?

**Reply:** accepted and deleted in order to avoid the confusion. This only refers to the initial stage of GWs. Since the pressure perturbations induced by GWs are relatively small in the initial stage, the overall pressure perturbations are not as obvious as that in the late stage.

- L. 310-313

As mentioned above, it is impossible to tell if it is downward-propagating without looking at the t-z section.

**Reply:** answered and explained in the first part.

- L. 320-328

Why can you say that these temperature and vertical wind perturbations were enhanced by GWs?

**Reply:** GWs are waves with energy and momentum. Weaker GWs induce weaker perturbations in temperature and vertical wind. Since the GWs are strengthened by the continuous descending of graupel/hail, the corresponding perturbations should be enhanced.

- L. 334-336

I do not understand which part of the figure you are referring to as upward-propagating GWs.

**Reply:** answered and explained in the first part.

- L. 344-345

Why do you think that it is due to the effect of upward-propagating GWs?

**Reply:** The region with negative temperature perturbations are now controlled by the upward

GWs, so we think that the weak in temperature perturbations are induced by the upward GWs. We cannot find other reasons to explain this phenomenon.

- L. 347-348

Is the wavelike structure of vertical velocity different from the above-mentioned GWs?

**Reply:** Yes. When the upward propagating GWs enter the stratosphere, they are trapped as energy and momentum and generate the horizontal propagating GWs with larger amplitude and lower frequency, which can propagate more distance in the stratosphere and cause potential influence on the atmospheric circulation in the stratosphere, such as QBO.

- L. 363-367

Why does continuous descending excite gravity waves, and why do GWs split storms?

**Reply:** This is related to the generation mechanism for the GWs in this study. As a trigger mechanism of the GWs proposed in this study, the descending of graupel/hail forces the air parcel to be displaced downward and break the air equilibrium state in a stable atmosphere and a recovery force of buoyancy is produced to act against it. The continuous descending of graupel/hail may excite buoyancy oscillations, which excites the GWs. The downward GWs will produce strong downdraft that split the main updraft of the storm rapidly.

- L. 367-369

In which part of the figures are GW amplitudes and wavelengths shown?

**Reply:** These values are estimated by calculating the distance between two peaks in pressure and vertical velocity perturbations. The wavelength is the distance between two neighboring peak pressure perturbations and amplitude is the maximum updraft perturbation in the initial stage and in the later stage for the GWs.

- L. 387-389

There does not appear to be any indication that GWs cause storms to split.

**Reply:** Fig.2 above clearly show that the main updraft in the storm is split by the downward GWs since the downdraft is caused by the GWs as answered above. We understand that this confusion might be caused by the unobvious splitting in total mixing ratio of hydrometeor. We change the mixing ratio as the simulated reflectivity, so the reflectivity splitting become much more obvious as shown in Fig.7.

- L. 398-408

Pressure and temperature perturbations have different structures. They do not look like due to the same GW.

**Reply:** Right. The obvious pressure perturbation structure is caused by downward GWs. However, the prominent temperature perturbations occurred in the stratosphere are caused by the trapped upward propagating GWs. When the upward propagating GWs enter the stable stratosphere, they are trapped and excite horizontal propagating GWs with larger wavelength and lower frequency.

- L. 434-443

I do not see a gravity wave structure in the figure. If there is also energy and momentum transport,

it should be shown in the figure.

   **Reply:** This part is deleted in order to avoid further confusion. Both the temperature and vertical velocity perturbations clearly show a wavelike propagation property. However, this is not a new finding in this paper and has already discussed in previous publications.

Figures
- Which altitude range is the hodograph in Fig. 1?
Reply: a fully hodograph is added as Fig.1b. The altitude range is numbered on the figure.

- The subscripts in Figs. 2 and 4 are missing.
**Reply:** accepted and revised.

- What does "composite" mean in Fig. 2? Does it mean integrated in altitude? If not, which altitude is drawn?
**Reply:** The composite reflectivity is the maximum dBZ reflectivity from any of the reflectivity angles of weather radar at every range, which is used to reveal the highest reflectivity in all echoes.

- The latitude of xz-section on the right side of Fig. 2 should be given by a line on the left side.
**Reply:** accepted and revised. The latitude information is added in the caption.

- The longitude ranges shown in Figs. 2a2-e2 should be the same.
**Reply:** the longitude ranges cannot be kept the same since the storm is moving. If using a large fixed range of longitude, the detailed structure is not clear.

- The contour labels in Figs. 3-5 are too small to read.
**Reply:** accepted and revised.

- How were the environmental positive and negative temperatures obtained? Deviation from initial values?
**Reply:** Yes. Some descriptions are added.

- Why are the figures arranged differently in Figs. 4 and 5?
**Reply:** accepted and revised. The different arrangement of figures in Fig.5 is only caused by that it is too small to arrange all figures in one panel. The figures after 30 min are deleted.

L. 175-176, 183-184
No southeastward extension is seen in Figs. 2b1 and 2c1.
**Reply:** accepted and revised. The features mainly occur in the late stage of the storm.

L. 178-181
Please cite references that show a relationship between the magnitude of reflectivity and graupel/hail loading.
**Reply:** accepted and added.

L. 239-240

"All modeled features are well consistent …" is an exaggeration. It is already split in the observation, but is not seen in Fig. 3c. Should be a correct description of what is consistent and what is not.

**Reply:** accepted and revised.

L. 285

"Perturbation" is perturbation from what? From the initial value? Explicitly state it.

**Reply:** yes, all perturbations are relative to initial state values. The relevant descriptions are added.

L. 306-310

Why does a collapse of equilibrium cause a strong restoring force of buoyancy? Does it mean that the drag of the falling particles pulls on the surrounding air and the restoring force acts against it?

**Reply:** yes, you are right. The detailed explanations are given in the answer to - L. 363-367

L. 433-434

I think that the cooled lower layer stabilize and do not rise.

**Reply:** accepted and revised. This sentence is unclear and revised as "the strong cold pool spreading causes the surrounding moist warm air to lift and condense", which is main mechanism for the effect of cold pool on the subsequent convection development.

Technical corrections:

L. 81 and many places

Please replay "stratospheric atmosphere" by "stratosphere".

**Reply:** accepted and revised.

L. 143 and many places

Please add "BST" after the time expression.

**Reply:** accepted and revised.

L. 273

Are graupel/hail mixing ratio and total hydrometeor mixing ratio the same or different? If they are the same, the same expression should be used.

**Reply:** They are quite different. Graupel/hail mixing ratio is just part of total hydrometeor mixing ratio which includes cloud ice, cloud water, rain water, snow and graupel/hail.

Additional descriptions are added in the revised manuscript.

L. 319

Which of vertical or horizontal does the wavelength mean?

**Reply:** As explained above, this is roughly estimated by calculating the distance between two peaks in the pressure and vertical velocity perturbations based on Fig. 4b$_2$ and 4d$_2$. More details see the reply to - L. 367-369.

---

## Author Comment (AC2)

**Reply to comments from referee 2:**

Comment on acp-2022-559
Anonymous Referee #2
This paper presents results from a single mesoscale model simulation of a thunderstorm. In its present form, I do not think the paper sufficiently advances the state-of-the-art to warrant publication. My reasons are as follows.
**Reply:** Thank a lot for your important comments. We carefully consider all comments and reply as following.

■ The model seems to be over two decades old. In the late '90s, when many of the referenced articles were written, 3D mesoscale models had 1 to 2 km horizontal grids, 0.5 km vertical grids, 100 to 500 km horizontal domains, and vertical domains reaching the Stratopause. This model seems to belong to that family with a 35km horizontal domain. By comparison, the 2018 Muller et al. paper looks at convection-allowing simulations with a 5000 km horizontal domain.

   **Reply:** The main result in our study find that the upper-level high loading of graupel/hail can generate downward propagating gravity waves when descending rather than thermal or mechanical processes. It means that the model used for this purpose must have an ability to simulate hail and hailstorm in details.

   Hail and hailstorms simulations are not available in most GCM models or climate models owing to that the inclusion of hail process in models not only require the high resolution but also need relevant physical processes. The very high terminal velocity for hail particles always causes stability problems. In our paper we use a hail-bin microphysics rather than hail parameterization scheme as used in most previous storm-scale models in order to appropriately simulate the hail falling process and associated gravity waves. For this purpose, the storm-scale high-resolution cloud models with detailed hail processes are the best choice for theoretically interpret the observed phenomenon.

   Muller et al. (2018) conducted many sensitivity experiments to resolution for convection-allowing simulations, however, cloud water, cloud ice, snow and rainwater processes are included in their models but no hail process (Stevens et al., 2013; Satoh et al., 2014). Therefore, these models can be used for thermally or mechanically induced gravity waves in convection, and cannot be used for gravity waves generated by hailstorms as this study.

■ It is not clear to me how the authors can confidently ascribe the downward propagating gravity waves to the novel process since the "buoyancy restoration force" occurs in the same area where the updraft overshoots the tropopause. I would have expected the authors to conduct a spectral analysis of the downward propagating gravity waves in order to identify clear distinguishing spectral properties (vertical and horizontal wavelengths and frequency) to associate with the length scales of the suggested originating process. The authors claim that it is necessary to understand these new waves because of the role they play in tropospheric dynamics. I do not see where the authors make the case for an important role for downward propagating waves. The only argument I discern is that these waves cause storm splitting. But storm splitting by downward propagating waves is argued based on the fact that the split occurs

at a given time. This explanation is unsatisfying. Storm splitting is a common phenomenon. Is it always caused by downward propagating waves?

**Reply:** The main reason to ascribe the downward propagating gravity waves reported in this study to a novel process is that the downward gravity waves are generated by the hail process rather than thermal or mechanical forcing although the "buoyancy restoration force" induced by the descending of graupel/hail is similar to those induced by thermal and mechanical forcing (Fig.1).

The upward propagating gravity waves are also generated by the storm top in the development stage for our simulated storm as reported in previous studies (Fig.2), however, the downward gravity waves generated by hail process occurs in the mature and decaying stages and the generation mechanisms are completely different from those found in previous studies.

To date, we found that the important role for the downward propagating gravity waves can cause the storm splitting rapidly (Fig.3), the issue is very important to the storm tracking and forecasting since the severe storms always cause significant damages to the public property. As you said, the storm splitting is a common phenomenon. The mechanisms that cause the storm splitting have been intensively investigated. The main mechanisms can be attributed to two aspects, one is related to interactions among wind shear, pressure perturbation and updraft development. The other is related to the precipitating-induced downdraft. We indicate that downward gravity waves generated by severe overshooting storm can be critical to storm splitting. However, issue relevant to storm splitting is not a main topic of this study.

For your suggestions to conduct spectral analysis in the downward propagating gravity waves, we will consider carefully. This study just physically interprets the generation process for gravity waves induced by a hailstorm and their potential impacts. The wave properties such as wave lengthen, duration and amplitude are estimated and found to be generally consistent with those found by previous studies.

■ As far as the upward propagating waves caused by reflection from the surface go, the authors claim that they "significantly change the dynamic and thermodynamic structure in the lower stratosphere". I do not see that a significant effect was measured or even described. Did the waves break and deposit momentum?

**Reply:** This phenomenon can be seen clearly when upward gravity waves reflected by the surface enter the stratosphere and induce strong fluctuations in temperature and vertical velocity (Fig.4-6). As you said, when upward gravity waves enter the stable stratosphere and they will deposit momentum and induce strong perturbations in temperature and vertical velocity, showing horizontal propagating gravity waves in the layer, and then breaking and decaying.

■ Perhaps the authors could consider extending the physical and temporal domain of the simulation and produce a spectral analysis of the waves they detect in order to support their conclusions that a new generating process is being observed. They should also produce quantitative arguments that downward propagating GWs cause storm splitting, and that

ground-reflected GWs have a significant effect on stratospheric dynamics.

**Reply:** Thanks a lot for this final comment. As stated above, the storm-scale storm model with hail-bin microphysics is an appropriate choice to simulate the gravity waves generated by the upper-level high graupel/hail loading. We will further revise and improve our manuscript based on your important comments.

[Figure]

Fig.1 Downward propagating waves induced by the descending of upper-level high graupel/hail loading.

[Figure]

Fig.2 Upward propagating gravity waves induced by the surface reflection process

[Figure]

Fig.3 Storm updraft splitting induced by the downward propagating waves.

[Figure]

Fig.4 Temperature perturbations induced by upward propagating gravity waves in the lower stratosphere.

[Figure]

Fig. 5 Temporal-height distribution of (a) maximum updraft (m/s) and (b) downdraft (m/s) for the simulated storm, indicating that downward propagating gravity waves occur at first (a), and then a strong upward propagating waves are formed (b).

[Figure]

[Figure]

Fig.6. Temporal-height (z-t) distributions of (a) maximum temperature (℃) and (b) minimum temperature (℃) for the simulated storm on 19 June 2017, showing that the upward propagating gravity waves deposit momentum in the stratosphere and induce a significant fluctuation in temperature in this layer.